# MM-LDM: Multi-Modal Latent Diffusion Model for Sounding Video Generation

## Abstract

Sounding video generation (SVG) is a challenging audio-video joint generation task that requires both single-modal realism and cross-modal consistency. Previous diffusion-based methods tackled SVG within the original signal space, resulting in a huge computation burden. In this paper, we introduce a novel multi-modal latent diffusion model (MM-LDM), which establishes a perceptual latent space that is perceptually equivalent to the original audio-video signal space but drastically reduces computational complexity. We unify the representation of audio and video signals and construct a shared high-level semantic feature space to bridge the information gap between audio and video modalities. Furthermore, by utilizing cross-modal sampling guidance, we successfully extend our generative models to audio-to-video and video-to-audio conditional generation tasks. We obtain the new state-of-the-art results with significant quality and efficiency gains. In particular, our method achieves an overall improvement in all evaluation metrics and a faster training and sampling speed [1].

## 1 Introduction

Sound Video Generation (SVG) is an emerging task in the field of multi-modal generation, which aims to integrate auditory and visual signals for audio-video joint generation. This integrated sounding video closely simulates real-life video formats, providing immersive audiovisual narratives. The potential applications of SVG span multiple fields, including artistic creation, film production, game development, virtual reality, and augmented reality, making it an area worth exploring in depth.

Compared to single-modal tasks like audio or video generation, SVG is more challenging since it requires a deep understanding of the complex interactions between sound and visual content to ensure audio-video cross-modal consistency. To be specific, generating a high-quality sounding video requires depicting vivid object appearance and coherent movements, accompanied by realistic audio that aligns with the video content. Two key factors exacerbate this challenge. First, both video and audio are high-dimensional data, making it difficult to ensure the generation realism of both modalities with limited computational resources. Second, cross-modal content consistency is difficult to achieve due to the inherent differences in data representation and content information between video and audio modalities. Specifically, in terms of data representation, videos are 3D visual signals with RGB three color channels, whereas audios are 1D continuous auditory signals with a single amplitude channel. From the perspective of content information, videos capture dense visual dynamics that change over time, while audio captures sound waves made by various visible or invisible sources. These inherent differences significantly increase the difficulty of obtaining cross-modal consistency.

Inspired by the success of single-modal diffusion models (Villegas et al., 2023; Huang et al., 2023), recent works have been proposed for SVG using the diffusion model. MM-Diffusion (Ruan et al., 2023) introduced a multi-modal diffusion model designed to jointly generate audio and video, effectively capturing their joint distribution in their raw signal space. However, due to the high dimensionality of audio and video signals, this method demands substantial computational resources, limiting its ability to produce high-resolution videos. Moreover, MM-Diffusion employs a small cross-attention window with size no more than 8 for efficient calculation, which comes at the cost of sacrificing the cross-modal consistency to some extent.

---

[1]Our codes will be released if accepted.

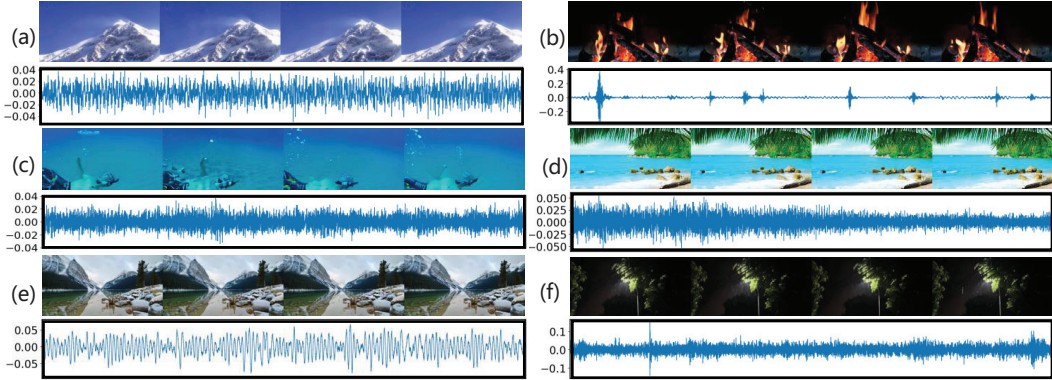

Figure 1: Sounding videos generated by our MM-LDM on the Landscape dataset (Lee et al., 2022). We can observe vivid scene like (a) mountain, (c) diving man, (e) lake, and so on. Matched audios are given like the sound of (b) wood burning, (d) sea wave, (f) raining, and so on. All presented audios can be played in Adobe Acrobat by clicking corresponding wave figures. More playable sounding video samples can be found in `https://anonymouss765.github.io/MM-LDM`.

To address the above challenges, we propose a novel Multi-Modal Latent Diffusion Model (MM-LDM) for SVG. **First**, we introduce a two-stage latent diffusion method to synthesize realistic audio and video signals with limited computational resources, In the first stage, we employ a multi-modal auto-encoder to map signals of each modality into a modal-specific perceptual latent space. These latent spaces are perceptually equivalent to the original signal spaces but significantly reduce computational complexity. In the second stage, a transformer-based diffusion model is adopted to perform SVG within the latent spaces. **Second**, to overcome the inherent differences between audio and video modalities, which pose the largest obstacles to achieving cross-modal consistency, we unify their data representations, allowing models to model their generation in a similar manner with shared parameters. Besides, since audio and video convey different messages (i.e., auditory and visual) even with a unified representation, we establish a shared high-level semantic feature space to bridge the gap of content information between the two modalities, thereby obtaining more valuable cross-modal insights. During training, we introduce a classification loss and a contrastive loss to optimize the high-level semantic feature space. Moreover, we can expand the applicability of our learned cross-modal correlations using cross-modal sampling guidance, allowing our generative diffusion model to perform cross-modal conditional generation tasks, including audio-to-video and video-to-audio generation. As shown in Fig. 1, MM-LDM can synthesize high-resolution ($256^2$) sounding videos with vivid objects, realistic scenes, coherent motions, and aligned audios. We conduct extensive experiments on the Landscape and AIST++ datasets, achieving new state-of-the-art generation performance with significant visual and auditory gains. For example, on the AIST++ dataset with $256^2$ spatial resolution, our MM-LDM outperforms MM-Diffusion by 114.6 FVD, 21.2 KVD, and 2.1 FAD. We also reduce substantial computational complexity, achieving a 10x faster sampling speed and allowing a larger sampling batch size.

**Our contributions:** 1) We propose a novel multi-modal latent diffusion model that establishes low-dimensional audio and video latent spaces for SVG, which are perceptually equivalent to the original signal spaces but significantly reduce the computational complexity. 2) We derive a shared high-level semantic feature space from the low-level perceptual latent spaces to provide cross-modal information that is easier to use when decoding audio-video signals. 3) By utilizing cross-modal sampling guidance, we successfully extend our SVG model to conditional generation tasks like audio-to-video and video-to-audio generation. 4) Our method obtains new state-of-the-art results on two benchmarks with significant audio and visual gains.

## 2 RELATED WORK

**Sounding Video Generation** SVG is a challenging multi-modal generation task since it requires: 1) synthesizing realistic high-dimensional video and audio signals with limited computational resources, and 2) bridging the representation and information gap between video and audio modalities to obtain content consistency. Several research works have been proposed to explore the challenging task of SVG. Based on generative adversarial networks, HMMD (Kurmi et al., 2021) introduces mul-

Figure 2: Overall illustration of our multi-modal latent diffusion model (MM-LDM) framework. Modules with **gray** border consist of our multi-modal autoencoder, which compresses raw audio-video signals into perceptual latent spaces. The module with **orange** border is our transformer-based diffusion model that performs SVG in the latent space. The **green** rectangle depicts the modification of inputs for unconditional audio-video generation (i.e. SVG), audio-to-video generation, and video-to-audio generation, respectively.

tiple discriminators to guide the generator in producing sounding videos with aligned audio-video content. Based on sequential generative models, UVG (Liu et al., 2023b) introduces a multi-modal tokenizer to encode different modal signals into discrete tokens, and employs a Transformer-based generator for SVG. Although these works have tackled the SVG task to some extent, their performances are far from expectations. Recently, motivated by the remarkable success of diffusion models (Ho et al., 2020; Song et al., 2021), MM-Diffusion (Ruan et al., 2023) is introduced to address SVG in the signal space. To ensure cross-modal consistency, MM-Diffusion introduces a random-shift method for efficient attention. However, this method suffers from a huge computational burden and uses a limited attention window size (typically no more than 8), resulting in suboptimal cross-modal consistency. In this paper, we propose MM-LDM that establishes a distinct low-level latent space for each modality and a shared high-level feature space for both modalities. The former spaces closely approximate the signal spaces while significantly reducing computational complexity. Simultaneously, the latter space provides valuable cross-modal information to obtain better cross-modal consistency.

**Latent Diffusion Model** Given that raw signal spaces for image, audio, and video modalities are of high dimensions, extensive efforts have been devoted to modeling their generation using latent diffusion models (Rombach et al., 2022; Liu et al., 2023a; Blattmann et al., 2023; Yu et al., 2023; He et al., 2022). For the image modality, LDM (Rombach et al., 2022) is devised to construct a perceptual latent space for images. This approach employs a KL-VAE to encode images into image latents and utilizes a latent diffusion model for text-to-image generation within the latent space. For the audio modality, AudioLDM (Liu et al., 2023a) is introduced to facilitate text-to-audio generation in a 1D latent space. In particular, it utilizes a large text-audio pretrained model CLAP (Wu et al., 2023) for extracting text and audio latent features. For the video modality, VideoLDM (Blattmann et al., 2023) is proposed to extend the LDM (Rombach et al., 2022) to high-resolution video generation. It introduces a temporal dimension into the LDM, and only optimizes these temporal layers while maintaining fixed, pretrained spatial layers. Despite previous latent diffusion models have demonstrated excellent performance on single-modal generation tasks, multi-modal generation tasks raise higher requirements, including unifying data representation across different modalities and considering cross-modal correlations. Our proposed method introduces a multi-modal latent diffusion model that synthesizes high-quality sounding videos with consistent audio-video content. It achieves new SoTA results on multiple benchmarks and significantly enhances the computational efficiency.

## 3 METHOD

In this section, we present our multi-modal latent diffusion model (MM-LDM) in detail. This approach consists of two main components: a multi-modal autoencoder designed for the compression of video and audio signals, and a multi-modal latent diffusion model for modeling SVG within latent spaces. Cross-modal sampling guidance is used to extend our method to cross-modal conditional generation tasks, including audio-to-video generation and video-to-audio generation tasks. A comprehensive overview of MM-LDM is illustrated in Fig. 2

### 3.1 VIDEO AND AUDIO COMPRESSION VIA A MULTI-MODAL AUTOENCODER

The detailed structure of our multi-modal autoencoder is presented in Fig. 3, which is composed of two modal-specific encoders, two signal decoders with shared parameters, two projector functions

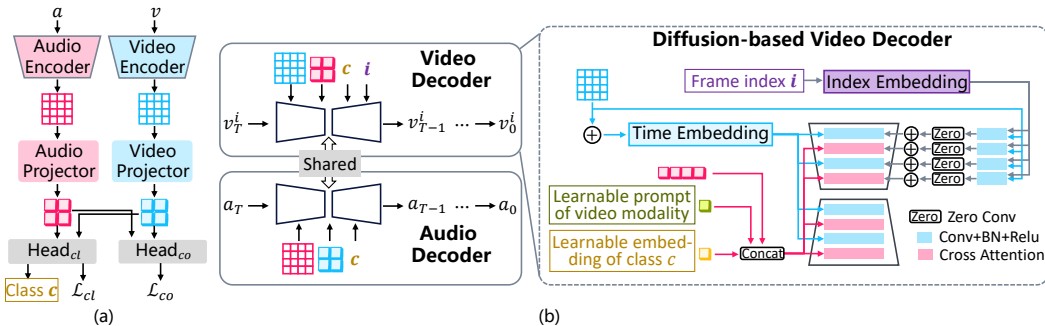

Figure 3: The detailed architecture of our multi-modal autoencoder. (a) Given a pair of audio and video inputs, two modal-specific encoders learn their perceptual latents. Two projectors map from two respective perceptual latent space to the shared semantic space. $\mathcal{L}_{cl}$ represents the classification loss and $\mathcal{L}_{co}$ denotes the contrastive loss. Class information can be obtained from the classification head during inference. (b) We share the decoder parameters and incorporate multiple conditional information for signal decoding. For the video modal, we provide a specific input of frame index to extract information of the target video frame.

mapping from each perceptual latent space to the shared semantic feature space, and two heads for classification and contrastive learning, respectively.

**Unifying Representation of Video and Audio Signals** We employ the raw video signals $v$ and transformed audio images $a$ to be our inputs. Video $v \in \mathbb{R}^{F \times 3 \times H \times W}$ can be viewed as a sequence of 2D images (i.e. video frames), where $F$, 3, $H$, and $W$ are frame number, channels, height, and width, respectively. Given that raw audio signals are 1D-long continuous data, we transform raw audio signals into 2D audio images to unify the representation of audio and video inputs. In particular, given raw audio signals, we first obtain its Mel Spectrogram with values normalized, which is denoted as $a_{raw} \in \mathbb{R}^{D \times T}$, where $D$ represents the number of audio channels and $T$ is the temporal dimension. Then, we treat $a_{raw}$ as a grayscale image and convert it into an RGB image using the PIL Python toolkit. Finally, we resize the Mel Spectrogram image to the same spatial resolution as the video input, obtaining an audio image $a \in \mathbb{R}^{3 \times H \times W}$.

Transforming raw audio signals into audio images is important for two key reasons. Firstly, as discussed in prior work (Huang et al., 2023), dealing with long continuous raw audio signals involves expansive computation complexity. Second, by unifying audio and video inputs into a shared image representation, we can leverage pre-trained image diffusion models to be our signal decoders, which not only reduces training consumption but also enhances decoding performance.

**Encoding Inputs into 2D Perceptual Latent Space** Given a pair of audio and video inputs, we employ a pretrained KL-VAE (Rombach et al., 2022) to downsample the video frames and audio image by a factor of $f$. Then, as depicted in Fig. 3(a), we introduce an audio encoder to compress the audio image into an audio perceptual latent $z_a \in \mathbb{R}^{C \times H_a \times W_a}$, which further downsamples the audio image by a factor of $f_a$, where $C$ is the number of channels, $H_a$ and $W_a$ denotes $\frac{H}{f \times f_a}$ and $\frac{W}{f \times f_a}$, respectively. Similarly, the video encoder compresses video frames into $z_v \in \mathbb{R}^{C \times H_v \times W_v}$.

The audio encoder is constructed in a similar way to the encoder part of U-Net, which consists of residual blocks and spatial attentions. Since video signals are temporally redundant (Sun et al., 2023b), we uniformly select keyframes from the input video to feed our video encoder. The structure of the video encoder differs from the audio encoder in two key aspects. Firstly, it adds a temporal attention layer after each spatial attention layer to capture temporal relationships. Secondly, an additional temporal pooling layer is employed before the final layer to integrate temporal information.

**Mapping Perceptual Latents to High-level Semantic Space** When decoding video and audio latents using a shared decoder, cross-model information is required to maintain consistency between these two modalities. In our experiments, we observed a significant performance drop when we directly used one perceptual latent as condition input to provide cross-modal information when decoding another perceptual latent. This performance drop reveals that the signal decoder is hard to extract useful cross-modal information from perceptual latents. This can be attributed to two key factors. Firstly, perceptual latents are dense representation of low-level information, thus presenting challenges for the decoder to comprehend. Secondly, video frames and audio images exhibit distinct

patterns - the former encompassing real-world object images while the latter are typical images in physics, thus the gap between their perceptual latents is too large for the decoder to bridge. To feed the video decoder with useful cross-modal information, specialized modules are required to extract high-level information from perceptual latents and narrow the gap between video and audio features.

To this end, as depicted in Fig. 3(a), we introduce an audio projector and a video projector that establish a shared high-level semantic space based on the low-level perceptual latents. In particular, the audio and video projectors extract semantic audio and video features $s_a$ and $s_v$ from their perceptual latents. To ensure the extracted features are of high-level semantic information, we employ a classification head $f_{cl}$ that takes a pair of audio and video features as inputs and predicts its class label, which is optimized using a classification cross-entropy loss. A contrastive loss is employed with a specified contrastive head to bridge the gap between video and audio features. The contrastive head $f_{co}$ maps $s_a$ and $s_v$ to 1D features respectively and calculates their contrastive loss with matched pairs of audio-video features being positive samples and all unmatched pairs of audio-video features as negative samples. Following (Radford et al., 2021), we define the contrastive loss as follows:

$$
\begin{aligned}
\mathcal{L}_{co} = &-\frac{1}{2}\sum_{i=1}^{B}\log\frac{\exp(\tau\cdot sim(f_{co}(s_a^i), f_{co}(s_v^i)))}{\sum_{j=1}^{B}\exp(\tau\cdot sim(f_{co}(s_a^i), f_{co}(s_v^j)))} \\
&-\frac{1}{2}\sum_{i=1}^{B}\log\frac{\exp(\tau\cdot sim(f_{co}(s_v^i), f_{co}(s_a^i)))}{\sum_{j=1}^{B}\exp(\tau\cdot sim(f_{co}(s_v^i), f_{co}(s_a^j)))}
\end{aligned}
\tag{1}
$$

where $sim(*)$ calculates the dot product of input features, $B$ and $\tau$ denote the batch size and a learnable parameter, respectively.

**Signal Decoding** As illustrated in Fig. 3(b), when reconstructing video signals, the signal decoder takes multiple factors into account, including the video perceptual latent $z_v$, frame index $i$, audio semantic feature $s_a$, learnable modality embedding, and learnable class embedding. The video perceptual latent provides spatial information for the $i$-th video frame using residual blocks, and content information through pooling. The audio semantic features are sequentialized and concatenated with the learnable modality embedding as well as the class embedding. Then they are fed to cross-attention layers to provide rich conditional information. When dealing with audio reconstruction, the signal decoder employs similar inputs, except for the frame index. More detailed explanations are presented in the appendix. To reduce training time and enhance the quality of reconstruction, we initialize our signal decoder with parameters of a pretrained image diffusion model (Rombach et al., 2022) and open all parameters during training.

**Training Targets** Following (Ho et al., 2020), we utilize the $\epsilon$-prediction to optimize our signal decoder, which involves the noise mean square error loss $\mathcal{L}_{MSE}$. Since the audio and video perceptual latents will be modeled for generation using MM-LDM, which will be specified in Sec. 3.2, we incorporate additional KL losses $\mathcal{L}_{KL}^a$ and $\mathcal{L}_{KL}^v$ to punish the distributions of audio and video latents towards an isotropic Gaussian distribution, which are similar to (Rombach et al., 2022). Previous works have proven the effectiveness of adversarial loss in training single-modal autoencoders (Esser et al., 2021; Sun et al., 2023a). Here, we introduce a novel adversarial loss to improve the quality of reconstructed multi-modal signals in terms of both single-modal realism and multi-modal consistency. Following (Sun et al., 2023a), we first obtain a pair of decoded video frames $\langle\bar{v}_i, \bar{v}_j\rangle$ with $i < j$ and corresponding audio image $\bar{a}$. Then, for the optimization of the discriminator, we select $\langle a, v_i, v_j\rangle$ as the real sample and $\langle\bar{a}, \bar{v}_i, \bar{v}_j\rangle$, $\langle\bar{a}, \bar{v}_i, v_j\rangle$, $\langle\bar{a}, v_i, \bar{v}_j\rangle$, and $\langle\bar{a}, \bar{v}_j, \bar{v}_i\rangle$ to be the fake samples. $\langle\bar{a}, \bar{v}_i, \bar{v}_j\rangle$ is viewed as the real sample for our autoencoder. Our adversarial loss can be formulated as $\mathcal{L}_{GAN}^D$ for the discriminator and $\mathcal{L}_{GAN}^{AE}$ for our autoencoder:

$$
\begin{aligned}
\mathcal{L}_{GAN}^D = &\log(1 - \mathcal{D}(\langle a, v_i, v_j\rangle)) + \log\mathcal{D}(\langle\bar{a}, \bar{v}_i, \bar{v}_j\rangle) + \log\mathcal{D}(\langle\bar{a}, \bar{v}_i, v_j\rangle) \\
&+ \log\mathcal{D}(\langle\bar{a}, v_i, \bar{v}_j\rangle) + \log\mathcal{D}(\langle\bar{a}, \bar{v}_j, \bar{v}_i\rangle) \\
\mathcal{L}_{GAN}^{AE} = &\log(1 - \mathcal{D}(\langle\bar{a}, \bar{v}_i, \bar{v}_j\rangle))
\end{aligned}
\tag{2}
$$

Our discriminator is constructed by several spatio-temporal modules that consist of residual blocks and cross-modal full attentions. Our final training loss for the multi-modal autoencoder becomes:

$$
\mathcal{L}_{AE} = \mathcal{L}_{MSE} + \lambda_{cl}\mathcal{L}_{cl} + \lambda_{co}\mathcal{L}_{co} + \lambda_{kl}(\mathcal{L}_{KL}^a + \mathcal{L}_{KL}^v) + \lambda_{gan}\mathcal{L}_{GAN}^{AE}
\tag{3}
$$

where $\lambda_{cl}$, $\lambda_{co}$, $\lambda_{kl}$ and $\lambda_{gan}$ are predefined hyper-parameters. $\mathcal{L}_{cl}$ and $\mathcal{L}_{co}$ are the classification and contrastive losses, respectively.

## 3.2 MULTI-MODAL LATENT DIFFUSION MODEL

As illustrated in Fig. 2, our approach independently corrupts audio and video latents during the forward diffusion process, whereas in the reverse denoising diffusion process, we employ a unified model that jointly predicts noise for both modalities. In particular, during forward diffusion, we corrupt audio and video latents, which are denoted as $z_a^0$ and $z_v^0$ (i.e. $z_a$ and $z_v$), by $T$ steps using a shared transition kernel. For simplicity, we use $z^0$ to represent both $z_a^0$ and $z_v^0$ in the subsequent section. Following prior works (Ho et al., 2020; Ruan et al., 2023), we define the transition probabilities as follows:

$$q(z^t|z^{t-1}) = \mathcal{N}(z^t; \sqrt{1-\beta_t}z^{t-1}, \beta_t\mathbf{I}); \quad q(z^t|z^0) = \mathcal{N}(z^t; \sqrt{\bar{\alpha}_t}z^0, (1-\bar{\alpha}_t)\mathbf{I}) \tag{4}$$

where $\{\beta_t \in (0,1)\}_{t=1}^T$ is a set of shared hyper-parameters, $\alpha_t = 1 - \beta_t$, and $\bar{\alpha}_t = \prod_{i=1}^t \alpha_i$. Utilizing Eq. (4), we obtain corrupted latents $z^t$ at time step $t$ as follows:

$$z^t = \sqrt{\bar{\alpha}_t}z^0 + (1-\bar{\alpha}_t)n^t \tag{5}$$

where $n^t \sim \mathcal{N}(0, \mathbf{I})$ represents noise features $n_a^t$ and $n_v^t$ for $z_a^t$ and $z_v^t$ respectively. the reverse diffusion processes of audio and video latents $q(z^{t-1}|z^t, z^0)$ have theoretically traceable distributions: To capture correlations between audio and video modalities and ensure content consistency, we introduce a unified denoising diffusion model $\theta$. This model takes both corrupted audio and video latents $(z_a^t, z_v^t)$ as input and jointly predicts their noise features $(n_a^t, n_v^t)$. The reverse diffusion process of corrupted audio and video latents is formulated as:

$$q((z_a^{t-1}, z_v^{t-1})|(z_a^t, z_v^t)) = \mathcal{N}((z_a^{t-1}, z_v^{t-1})|\mu_\theta(z_a^t, z_v^t), \tilde{\beta}_t\mathbf{I}) \tag{6}$$

During training, we minimize the mean square error between the predicted and original noise features of matched audio-video pairs, known as $\epsilon$-prediction in the literature (Kingma et al., 2021):

$$\mathcal{L}_\theta = \frac{1}{2}\|\tilde{n}_\theta^a(z_a^t, z_v^t, t) - n_a^t\|_2 + \frac{1}{2}\|\tilde{n}_\theta^v(z_a^t, z_v^t, t) - n_v^t\|_2 \tag{7}$$

Here, $\tilde{n}_\theta^a(z_a^t, z_v^t, t)$ and $\tilde{n}_\theta^v(z_a^t, z_v^t, t)$ are the predicted audio and video noise features, respectively. Given that our audio and video latents $z_a$ and $z_v$ possess relatively small spatial resolution, we employ a Transformer-based diffusion model known as DiT (Peebles & Xie, 2022) as our backbone model. During training, we sequentialize audio and video latents and independently add positional embeddings (Vaswani et al., 2017) to each latent. Then, two learnable token embeddings, $[EOS_a]$ and $[EOS_v]$, are defined and inserted before the audio and video latents, respectively. Finally, audio and video latents are concatenated and fed to DiT for multi-modal generation.

## 3.3 CONDITIONAL GENERATION

Initially designed for class-conditioned image generation (Ho & Salimans, 2022), the classifier-free guidance has demonstrated its effectiveness in the more difficult text-to-image generation task (Saharia et al., 2022; Ramesh et al., 2022). Inspired by this success, we introduce the cross-modal sampling guidance that targets both audio-to-video and video-to-audio generation tasks. Our approach involves training the single MM-LDM to simultaneously learn three distributions: an unconditional distribution denoted as $\tilde{n}_\theta(z_a^t, z_v^t, t)$ and two conditional distributions represented as $\tilde{n}_\theta(z_v^t, t; z_a)$ and $\tilde{n}_\theta(z_a^t, t; z_v)$, corresponding to the SVG, audio-to-video and video-to-audio generation tasks respectively. To accomplish this, we incorporate a pair of null audio and video latents, defined as $(\mathbf{0}_a, \mathbf{0}_v)$ with $\mathbf{0}_a = 0$ and $\mathbf{0}_v = 0$. Then, the unconditional distribution $\tilde{n}_\theta(z_a^t, z_v^t, t)$ can be reformulated to be $\tilde{n}_\theta(z_a^t, z_v^t, t; \mathbf{0}_a, \mathbf{0}_v)$. The conditional distribution $\tilde{n}_\theta(z_v^t, t; z_a)$ can be reformed as $\tilde{n}_\theta(z_a^t, z_v^t, t; z_a, \mathbf{0}_v)$, where $z_a^t$ can be obtained directly given $z_a$ ant $t$ according to Eq. (5). Similarly, $\tilde{n}_\theta(z_a^t, t; z_v)$ is reformulated as $\tilde{n}_\theta(z_a^t, z_v^t, t; \mathbf{0}_a, z_v)$. As depicted in Fig. 2, the conditional inputs are added to the input latents after zero convolution layers (which are ignored in the figure for conciseness) to provide conditional information. We randomly select 5% training samples for each conditional generation task. Finally, taking audio-to-video generation as an example, we perform sampling utilizing the following linear combination of the conditional and unconditional noise predictions, defined as follows:

$$\bar{n}_\theta^v(z_v^t, t; z_a) = \phi \cdot \tilde{n}_\theta^v(z_a^t, z_v^t, t; z_a, \mathbf{0}_v) - (1-\phi) \cdot \tilde{n}_\theta^v(z_a^t, z_v^t, t; \mathbf{0}_a, \mathbf{0}_v) \tag{8}$$

where $\phi$ serves as a hyper-parameter that controls the intensity of the conditioning.

Table 1: Efficiency comparison with MM-Diffusion (Ruan et al., 2023) on a V100 32G GPU, which models SVG within the signal space. MBS denotes the Maximum Batch Size.

| Method | Resolution | Training | | Inference | |
|---|---|---|---|---|---|
| | | MBS | One Step Time | MBS | One Sample Time |
| MM-Diffusion | $64^2$ | 4 | 1.70s | 32 | 33.1s |
| MM-Diffusion | $128^2$ | 1 | 2.36s | 16 | 90.0s |
| MM-LDM (ours) | $\mathbf{256^2}$ | 9 | 0.46s | 4 | 70.7s |
| MM-LDM* (ours) | $\mathbf{256^2}$ | **12** | **0.38s** | **33** | **8.7s** |

## 4 EXPERIMENT

In this section, we first introduce our experimental setups and implementation details in Sec. 4.1. Then we present the quantitative and qualitative comparison between our MM-LDM and prior works in Sec. 4.2 and Sec. 4.3. Finally, we conduct ablation studies in Sec. 4.4.

### 4.1 EXPERIMENTAL SETUPS

**Dataset** Following (Ruan et al., 2023), we conduct experiments on two distinct and high-quality sounding video datasets, namely Landscape (Lee et al., 2022) and AIST++ (Li et al., 2021). The former comprises videos recording nature scenes and the latter encompasses videos recording street dances. More detailed introduction of each dataset is presented in the appendix.

**Baseline Models** We compare our MM-LDM against four state-of-the-art methods: (1) TATS (Ge et al., 2022): a sequential generative model for video generation; (2) DIGAN (Yu et al., 2022): a generative adversarial network for video generation; (3) Diffwave (Kong et al., 2021): a diffusion-based method for audio generation; (4) MM-Diffusion (Ruan et al., 2023): a diffusion-based method for sounding video generation. MM-Diffusion initially synthesizes sounding videos with $64^2$ resolution and then utilizes a Super-Resolution (SR) model to obtain $256^2$ resolution.

**Evaluation Metrics** For video evaluation, we follow previous settings (Ge et al., 2022; Yu et al., 2022; Ruan et al., 2023) that employ the Fréchet Video Distance (FVD) and Kernel Video Distance (KVD) metrics for video evaluation and Fréchet Audio Distance (FAD) for audio evaluation. Our implementation for these evaluations is based on the resources provided by (Ruan et al., 2023). Our MM-LDM synthesize all videos at a $256^2$ resolution. We resize the synthesized videos when testing the metrics in the $64^2$ resolution.

**Implementation Details** When training our multi-modal autoencoder, we utilize pretrained KL-VAE (Rombach et al., 2022) with the downsample factor being 8. Both video frames and audio images are resized to a $256^2$ resolution, and video clips have a fixed length of 16 frames ($F = 16$). The audio and video encoders use downsample factors of $f_a = 4$ and $f_v = 2$, yielding latents of spatial size $8^2$ and $16^2$, respectively. The number of latent channels is 16 for both modalities. The loss weights $\lambda_{cl}$, $\lambda_{co}$, and $\lambda_{gan}$ are 1e-1, 1e-1, and 1e-1, respectively. The loss weight $\lambda_{kl}$ is set as 1e-9 for Landscape and 1e-8 for AIST++. Further details can be found in the appendix.

### 4.2 QUANTITATIVE COMPARISON

**Efficiency** We quantitatively compared the efficiency of our MM-LDM and MM-Diffusion and presented the results in Table. 1. MM-LDM incorporates both the auto-encoder and the DiT generator, while MM-LDM* only tests the DiT generation performance by preprocessing and saving all latents. We fix the batch size being 2 when testing the one-step time during training and the one sample time during inference, and DDIM sampler is used with 50 steps for both methods.

Since MM-Diffusion operates in the signal space, it demands huge computational complexity when the spatial resolution of synthesized videos increases. In particular, it struggles to model high-resolution ($256^2$) video signals on a 32G V100, leading to the out-of-memory error. We evaluate the efficiency of MM-Diffusion with two spatial resolution settings: $64^2$ and $128^2$. MM-LDM demonstrates improved efficiency with higher video resolutions ($256^2$ vs. $128^2$) during the training process. When employing the same batch size (i.e., 2), our MM-LDM outperforms MM-Diffusion by 6x speed for each training step, allowing a much bigger training batch size with the higher video resolution. During inference, our diffusion generative model DiT, which performs SVG within the latent space, achieves a 10x faster sampling speed and allows a larger sampling batch size.

Table 2: Quantitaive comparison on the Landscape and AIST++ datasets for SVG, audio-to-video, and video-to-audio generation. Results noted with ∗ are reproduced with their released checkpoints.

| Method | Resolution | Sampler | Landscape | | | AIST++ | | |
|---|---|---|---|---|---|---|---|---|
| | | | FVD ↓ | KVD ↓ | FAD ↓ | FVD ↓ | KVD ↓ | FAD ↓ |
| Ground Truth | $64^2$ | - | 16.3 | -0.015 | 7.7 | 6.8 | -0.015 | 8.4 |
| Ground Truth | $256^2$ | - | 22.4 | 0.128 | 7.7 | 11.5 | 0.043 | 8.4 |
| *Single-Modal Generative Models* | | | | | | | | |
| DIGAN | $64^2$ | - | 305.4 | 19.6 | - | 119.5 | 35.8 | - |
| TATS-base | $64^2$ | - | 600.3 | 51.5 | - | 267.2 | 41.6 | - |
| MM-Diffusion-v | $64^2$ | dpm-solver | 238.3 | 15.1 | - | 184.5 | 33.9 | - |
| Diffwave | $64^2$ | - | - | - | 14.0 | - | - | 15.8 |
| MM-Diffusion-a | - | dpm-solver | - | - | 13.6 | - | - | 13.3 |
| *Multi-Modal Generative Models on Audio-to-Video Generation* | | | | | | | | |
| MM-Diffusion* | $64^2$ | dpm-solver | 237.9 | 13.9 | - | 163.1 | 28.9 | - |
| MM-Diffusion-SR* | $64^2$ | dpm-solver+DDIM | 225.4 | 13.3 | - | 142.9 | 24.9 | - |
| MM-LDM (ours) | $64^2$ | DDIM | **89.2** | **4.2** | - | **71.0** | **10.8** | - |
| MM-Diffusion-SR* | $256^2$ | dpm-solver+DDIM | 347.9 | 27.8 | - | 225.1 | 51.9 | - |
| MM-LDM (ours) | $256^2$ | DDIM | **123.1** | **10.4** | - | **128.5** | **33.2** | - |
| *Multi-Modal Generative Models on Video-to-Audio Generation* | | | | | | | | |
| MM-Diffusion* | - | dpm-solver | - | - | 9.6 | - | - | 12.6 |
| MM-LDM (ours) | - | DDIM | - | - | **9.2** | - | - | **10.2** |
| *Multi-Modal Generative Models on Sounding Video Generation* | | | | | | | | |
| MM-Diffusion | $64^2$ | DDPM | 117.2 | 5.8 | 10.7 | 75.7 | 11.5 | 10.7 |
| MM-Diffusion | $64^2$ | dpm-solver | 229.1 | 13.3 | 9.4 | 176.6 | 31.9 | 12.9 |
| MM-Diffusion+SR* | $64^2$ | dpm-solver+DDIM | 211.2 | 12.6 | 9.9 | 137.4 | 24.2 | 12.3 |
| MM-LDM (ours) | $64^2$ | DDIM | **77.4** | **3.2** | **9.1** | **55.9** | **8.2** | **10.2** |
| MM-Diffusion+SR* | $256^2$ | dpm-solver+DDIM | 332.1 | 26.6 | 9.9 | 219.6 | 49.1 | 12.3 |
| MM-LDM (ours) | $256^2$ | DDIM | **105.0** | **8.3** | **9.1** | **105.0** | **27.9** | **10.2** |

**Quality** We quantitatively compare our method with prior works for the audio-to-video generation, video-to-audio generation, and sounding video generation tasks. The results are reported in Table. 2. We first evaluate model performance on the single-modal conditional generation tasks like audio-to-video and video-to-audio generation. On the landscape dataset, our MM-LDM outperforms MM-Diffusion by 136.2 FVD and 55.9 KVD at the $64^2$ resolution, 224.8 FVD and 17.4 KVD at the $256^2$ resolution, and 0.4 FAD. On the AIST++ dataset, our MM-LDM outperforms MM-Diffusion by 71.9 FVD and 14.1 KVD at the $64^2$ resolution, 96.6 FVD and 18.7 KVD at the $256^2$ resolution, and 2.4 FAD. These results demonstrate that our MM-LDM captures more insightful cross-modal information and demonstrates the effectiveness of our cross-modal sampling guidance.

For the multi-model joint generation task (i.e., SVG) at the $64^2$ resolution, we achieve a 39.8 FVD, 2.6 KVD and 1.6 FAD improvement on the Landscape dataset and a 19.8 FVD, 3.3 KVD and 0.5 FAD improvement on the AIST++ dataset compared to MM-Diffusion. At the $256^2$ resolution, we achieve a 227.1 FVD, 18.3 KVD, and 0.8 FAD improvement on the Landscape dataset and a 114.6 FVD, 21.2 KVD and 2.1 FAD improvement on the AIST++ dataset. It can be seen that our method enhances the generation quality more significantly when the resolution increases, demonstrating the necessity of establishing perceptual latent spaces for high-resolution sounding video generation. Notably, when using the DDPM sampler, MM-Diffusion requires 1000 diffusion steps to synthesize a sounding video sample, taking approximately 8 minutes for a single sample. In contrast, our MM-LDM synthesizes higher-resolution videos with only 50 sampling steps using the DDIM sampler. Furthermore, our research reveals that cross-modal generation yields a substantial enhancement in the quality of each single modality, demonstrating the potential of multi-modal generation tasks.

## 4.3 QUALITATIVE COMPARISON

We qualitatively compare the generative performance of our MM-LDM and MM-Diffusion in Fig. 4, using the provided checkpoints for MM-Diffusion when sampling. All images are at a resolution of $256^2$. Videos synthesized by MM-Diffusion produce blurry appearances with deficient details, whereas our MM-LDM yields more clear samples with better audio-video alignments.

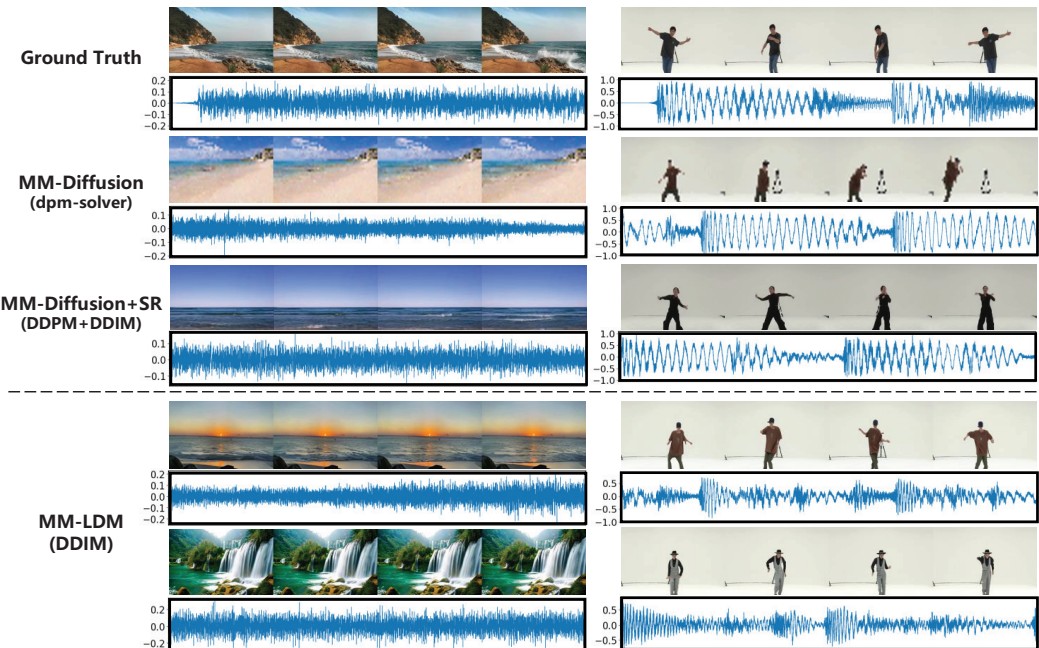

Figure 4: Qualitative comparison of sounding video samples: MM-Diffusion vs. MM-LDM (ours). All presented audios can be played in Adobe Acrobat by clicking corresponding wave figures.

Table 3: Ablation study on the multi-modal autoencoder on the Landscape dataset.

| Learnable Modality Prompt | Latent Average Pooling | Learnable Class Embedding | Semantic Cross-Modal Feature | Finetune KL-VAE Decoder | Adversarial Training Loss | rFVD ↓ | rKVD ↓ | rFAD ↓ |
|---|---|---|---|---|---|---|---|---|
| | | | | | | 110.7 | 6.9 | 9.3 |
| ✓ | | | | | | 105.5 | 6.8 | 9.0 |
| ✓ | ✓ | | | | | 94.4 | 5.7 | 9.2 |
| ✓ | ✓ | ✓ | | | | 87.7 | 5.1 | 9.1 |
| ✓ | ✓ | ✓ | ✓ | | | 80.1 | 4.3 | 9.1 |
| ✓ | ✓ | ✓ | ✓ | ✓ | | 75.7 | 3.9 | 8.9 |
| ✓ | ✓ | ✓ | ✓ | ✓ | ✓ | **53.9** | **2.4** | **8.9** |

## 4.4 ABLATION STUDIES

We conduct ablation studies on our multi-modal autoencoder and report the results in Table. 3. Our base autoencoder independently decodes signals for each modality based on the respective spatial information from perceptual latents. To reduce computational complexity, we share the diffusion-based signal decoder for both modalities while using a learnable embedding to prompt each modality, obtaining improved performance. For a more comprehensive content representation, we apply average pooling to each perceptual latent, adding the latent feature to the timestep embedding and further enhancing the model performance. By incorporating the classification and contrastive losses, we leverage prior knowledge of class labels and extract high-level cross-modal information, which significantly boosts model performance. Since the KL-VAE was originally trained on natural images and is unfamiliar with physical images like audio images, we finetune its decoder on each dataset for better performance. Finally, after training with the adversarial loss, our autoencoder attains its best reconstruction performance, achieving 53.9 rFVD, 2.4 rKVD, and 8.9 rFAD.

## 5 CONCLUSION

This paper introduces MM-LDM, a novel diffusion-based method for SVG. Our method reduces the computational complexity of SVG by constructing low-dimensional latent spaces that faithfully capture the perceptual characteristics of the original signal spaces. To obtain cross-modal consistency, we unify the audio-video data representation and bridge the gap of content information between the two modalities. Extensive experiments demonstrate that our method excels at generating sounding videos with enhanced single-modal realism and cross-modal consistency, achieving new state-of-the-art results on multiple benchmarks with better training and inference efficiency.

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
