# MM-LDM: Multi-Modal Latent Diffusion Model for Sounding Video Generation

## 1 Appendix

### 1.1 Limitations

Our research exhibits several limitations. First, when unifying the data representation of audio and video inputs, we resize the audio Mel Spectrogram image to the same spatial resolution with video frames, which may lead to information loss to some extent. Second, we utilize a shared KL-VAE and two modal-specific encoders to compress audio and video signals, which significantly reduces the computational complexity but can sacrifice fine-grained details. Third, while previous video generation studies have shown the promise of diffusion models in generating open-domain videos, we have not yet extended our method to open-domain sounding video datasets.

### 1.2 Signal Decoding

As illustrated in Fig. 3(b) in the main paper, when reconstructing video signals, the signal decoder takes multiple factors into account, including the video perceptual latent $z_v$, frame index $i$, audio semantic feature $s_a$, learnable modality embedding, and learnable class embedding. The video perceptual latent plays a vital role in guiding the video reconstruction in two ways. Firstly, it provides detailed spatial information for the $i$-th video frame by utilizing residual blocks. This spatial information is integrated into each residual block within the signal decoder following a zero convolution to provide spatial signal guidance, which is the most basic condition of our method. Secondly, we spatially average pool the video perceptual latent and add it with the timestep embedding to offer global content guidance. To incorporate cross-modal information, we sequentialize the audio semantic feature and feed it to the cross attention layers. Since we share signal decoder parameters for both audio and video modalities, we introduce learnable prompt embeddings specific to each modality for distinction. In addition, given that our classification head can predict class labels precisely, we further define a learnable embedding for each class to incorporate more prior knowledge. Both the modality embedding and the class embedding are concatenated with the audio semantic feature to feed cross-attention layers. When dealing with audio reconstruction, the signal decoder employs similar inputs, except for the frame index. To reduce training time and enhance the quality of reconstruction, we initialize our signal decoder with parameters of a pretrained image diffusion model (Rombach et al., 2022) and open all parameters during training.

### 1.3 Dataset Details

Following (Ruan et al., 2023), we conduct experiments on two distinct and high-quality sounding video datasets, namely Landscape (Lee et al., 2022) and AIST++ (Li et al., 2021). The Landscape dataset comprises 1,000 non-overlapping video clips recording nature scenes. Each video clip is 10 seconds, obtaining a total duration of about 2.7 hours and encompassing 300K frames. These videos are categorized into nine classes, each corresponding to a different nature scene. The AIST++ dataset is a subset of the AIST dataset (Tsuchida et al., 2019) and encompasses 1,020 video clips recording street dances. This dataset has a total duration of around 5.2 hours, containing a total of 560K frames. For a fair comparison, we utilize the center-cropped version of the AIST++ dataset provided by (Ruan et al., 2023).

## 1.4 MORE IMPLEMENTATION DETAILS

The multi-modal autoencoder is first trained without the adversarial loss for 30 epochs on the Landscape dataset and 10 epochs on the AIST++ dataset. Then, we introduce the adversarial loss and continue training for an additional 30 epochs on the Landscape dataset and 10 epochs on the AIST++ dataset. Notably, we stop the gradient of perceptual latents $z_a$ and $z_v$ when feed them to the projectors using $detach()$. For the training of the generator diffusion modal, i.e., DiT, we optimize it for 300 and 100 epochs on the Landscape and AIST++ datasets, respectively. Our methods are implemented using PyTorch (Paszke et al., 2019), and all experiments are conducted on 8 NVIDIA A100 GPUs. The detailed settings of model hyper parameters are presented in Table. 1.

Table 1: Hyper-parameters of the multi-modal auto-encoder and the DiT.

|  | Landscape | AIST++ |
|---|---|---|
| **KL-VAE** | | |
| $f$ | 8 | |
| **Video Encoder** | | |
| $f_a$ | 4 | |
| $f_v$ | 2 | |
| Input Shape | 32 | |
| Input Channels | 4 | |
| Output Channels | 16 | |
| Model Channels | 320 | |
| Num Res. Blocks | 2 | |
| Num Head Channels | 64 | |
| Attention Resolutions | [16, 8] | |
| Channel Multiplies | [1, 2] | |
| **Video Decoder (UNet)** | | |
| Input Shape | 32 | |
| Input Channels | 4 | |
| Output Channels | 4 | |
| Model Channels | 320 | |
| Num Res. Blocks | 2 | |
| Num Head | 8 | |
| Attention Resolutions | [32, 16, 8] | |
| Channel Multiplies | [1, 2, 4, 4] | |
| **Video Generator (DiT)** | | |
| Input Shape | 16 | |
| Input Channels | 16 | |
| Model Channels | 1152 | |
| Num Head | 16 | |
| Depth | 28 | |
| Mlp Ratio | 4 | |

## 1.5 OPEN-DOMAIN VIDEO GENERATION

To evaluate the performance of our proposed method for open-domain generation, we train it on the 10% training from AudioSet dataset for 5 days in total (2 days for the multi-modal autoencoder and 3 days for the multi-modal generator) with 2 A800 gpus. Samples are presented in Fig. 1. It can be seen that our method can synthesize diversity video content with consistent audios, demonstrating the effectiveness of our method on open-domain sounding video generation.

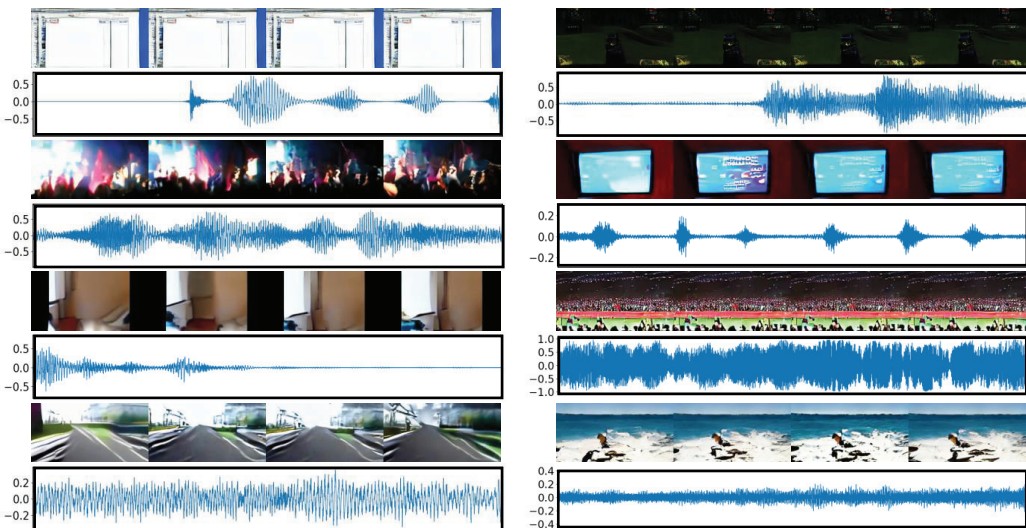

Figure 1: Samples of open-domain sounding video generation on the AudioSet dataset.

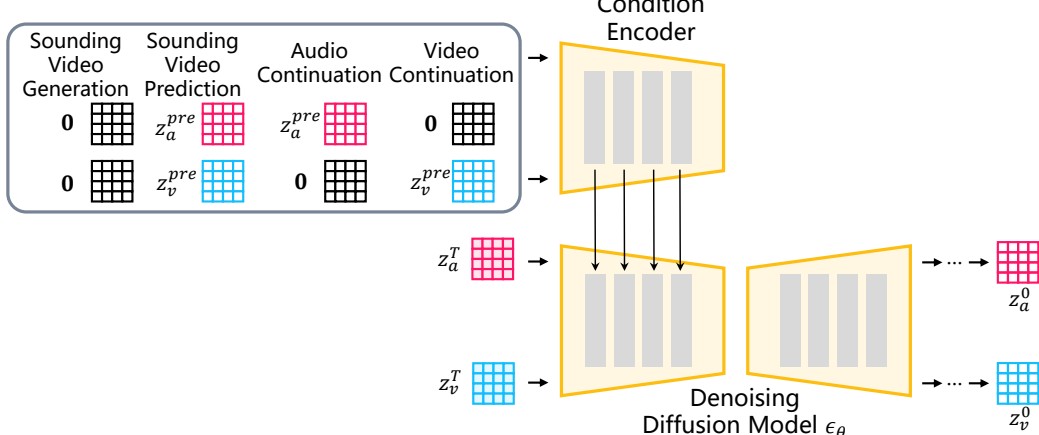

Figure 2: We extend our MM-LDM to long video generation by incorporating a condition encoder, thus enabling to model the sounding video generation, sounding video prediction, audio continuation, and video continuation simultaneously. $z_a^{pre}$ and $z_v^{pre}$ denotes the synthesized previous audio and video latents.

## 1.6 LONG VIDEO GENERATION

For long video generation, we introduce a condition encoder as shown in Fig. 2 and perform experiments on the AIST++ dataset. We consider four generation tasks: sounding video generation, sounding video prediction, audio continuation, and video continuation. The sounding video generation task takes non-conditions as input, synthesizing the initial sounding video clip. Then the sounding video predict task takes the synthesized previous sounding video clip as inputs, synthesizing the subsequent sounding video clip. By combining the two tasks, we can synthesize long sounding videos in an auto-regressive manner. We also incorporate the audio continuation task by replacing the video latent with the zero latent, and the video continuation task is integrated in a similar way. During each training iteration, we randomly select a task with uniform distribution. We also present samples of long video generation, audio continuation, and video continuation in Fig. 3. It can be seen that our method can synthesize consistent long sounding videos, demonstrating the effectiveness of our proposed method.

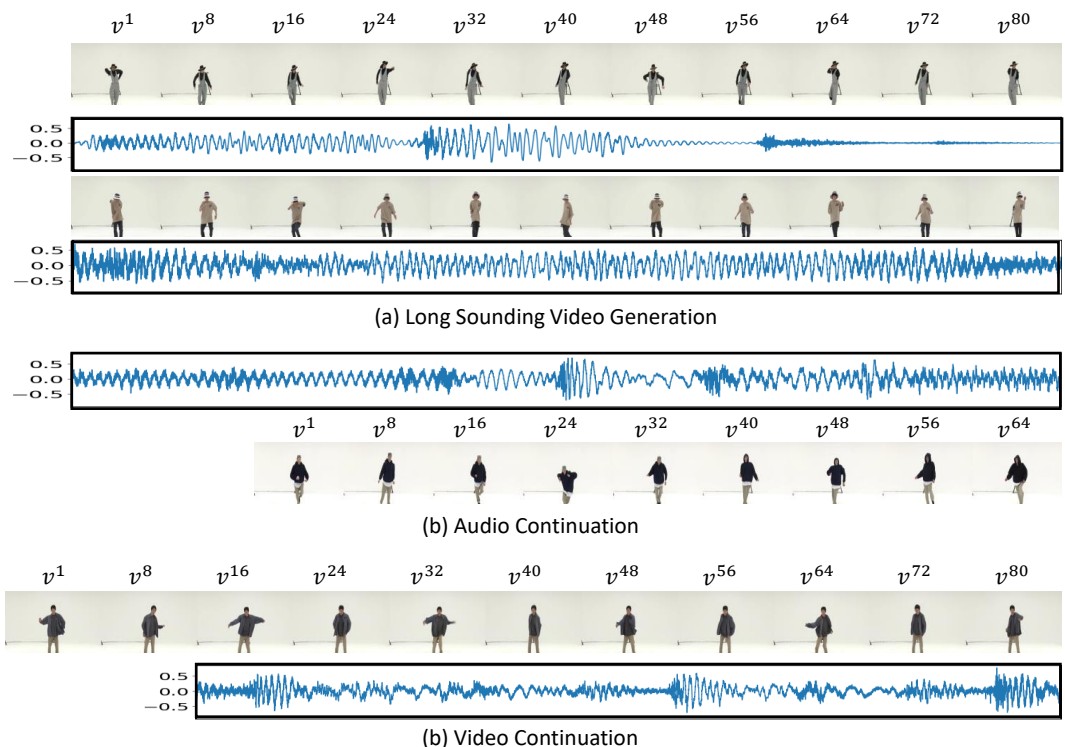

Figure 3: Samples of long video generation, audio continuation, and video continuation on the AIST++ dataset.