# OpenReview forum: "MM-LDM: Multi-Modal Latent Diffusion Model for Sounding Video Generation"
_ICLR.cc/2024/Conference — Submitted to ICLR 2024_

### Official Review · Reviewer_n6co · 2023-10-30

**Soundness:** 2 fair
**Presentation:** 2 fair
**Contribution:** 2 fair
**Rating:** 5
**Confidence:** 4

**Summary:**

The paper proposes a multi-modal latent diffusion model named SVG for audio and video generation. Both audio and video signals are into latent spaces and then learn joint semantic features via classification and contrastive loss. The resulting semantic features can be used as conditional signals to improve audio-to-video and video-to-audio generation. The experiments were conducted on two sounding video datasets and the quantitative results are better than the baselines.

**Strengths:**

1) Audio-visual cross-modal generation is a challenging task. The authors proposed a promising approach to bridge the gap between audio and video efficiently.
2) The quality of the generated samples looks good to me.
3) The paper is easy to understand.

**Weaknesses:**

1) The paper is more like a straightforward follow-up on MM-Diffusion. The major difference is changing the diffusion targets from the raw signal domains to the latent spaces. However, based on the literature, it is almost trivial or obvious that transferring to latent space could achieve better results and improve training efficiency in any diffusion generation. From this perspective, the technical novelty of this paper is limited.
2) The datasets used in the paper are pretty limited. The AIST and Landscape are small-size datasets. The proposed method could overfit the dataset, and indeed, diffusion is really good at overfitting. While the authors mentioned that they have not yet extended the method to open-domain sounding video datasets, I believe that is actually the critical research problem required to solve.
3) While the quality of the generated samples on the webpage is nice for 1 second, I believe it is quite limited and probably hard to generalize for a longer time.
4) In addition to quantitative metrics, I believe it is quite important to have a subjective evaluation of the generated samples, especially for audio.

**Questions:**

1) Have you tried doing applications on audio-visual continuation? For example, the model is conditioned on the first 1s audio, and then you can generate the next 1s audio and visual together. Based on your approach, it seems like these applications are also feasible.
2) There exist so many losses and learnable embeddings within the audio-visual autoencoder. How do you search those weights of different losses? While there are ablation studies, it is only in one direction. I am interested in how you eventually could find the best combinations of all terms.

---

> ### Author Response · Authors · 2023-11-17
> **Official Comments for Reviewer n6co**
>
> # Weakness #1: Method Innovation
> In addition to constructing latent spaces, another innovation of this paper is addressing the inherent differences between two modalities for consistent generation. Previous works on constructing latent space mostly focus on single-modal generation. In contrast to prior works, we have to consider a broader range of issues , such as unifying data representations and bridging the semantic gap between different modalities. In this work, we tackle these issues and achieve exciting cross-modal consistency. Our ablation studies also demonstrate the importance of establishing cross-modal correlations when constructing latent features for the multi-modal generation task.
>
> # Weakness #2: Open-domain Experiments
> Thank you for your suggestion. Please refer to the official comment Q1.
>
> # Weakness #3: Long Video Generation
> There is no need to violently increase the length of sounding video in our Multi-modal auto-encoder to synthesize long videos. Instead, we can synthesize long videos in an auto-regressive manner like prior works [1][2], which synthesize latents of  subsequent video clips based on that of previous synthesized video clips. In practice, to extend our MM-LDM to long video generation, we utilize a condition encoder that takes the previous sounding video clip as input to guide the MM-LDM to synthesize subsequent sounding video clips. We have presented our new proposed approach and long sounding video samples in Appendix 1.6, where all audios can be displayed using Adobe Acrobat PDF Reader by clicking the figures. Please refer to our new PDF for details.
>
> [1] Yu S, Sohn K, Kim S, et al. Video probabilistic diffusion models in projected latent space. CVPR 2023: 18456-18466.
> [2] He Y, Yang T, Zhang Y, et al. Latent video diffusion models for high-fidelity long video generation[J]. arXiv preprint arXiv:2211.13221, 2023.
>
> # Weakness #4: Human Evaluation
> Thank you for your comments. Please refer to the official comment Q2.
>
> # Question #1: Audio-visual Continuation
> Thank you for your comments! It is an interesting task and we incorporate it when extending our MM-LDM for long video generation. Please refer to Appendix 1.6 in our new PDF for details.
>
> # Question #2: Ablation Study on Loss Weights
> Thank you for your question. Following prior work [1], we consistently utilized an adversarial learning weight of 0.1 in all experiments. In addition, we experimented with four configurations of loss weights, each obtained after a 20-epoch run (approximately 16K steps) on the AIST++ dataset. Across different settings, we assess metrics such as FVD, KVD, and FAD for samples at resolutions of 64 and 256. As reported in the following table, the performance of configuration #3 outperforms that of other configurations at both resolutions.  Thus, we select configuration #3 as the final configuration used in the paper.
> [1] Sun M, Wang W, Zhu X, et al. MOSO: Decomposing MOtion, Scene and Object for Video. CVPR 2023.
> |#|$\lambda_{kl}$|$\lambda_{co}$|$\lambda_{cl}$|FVD 64x| KVD 64x|FAD 64x|FVD 256x|KVD 256x|FAD 256x|
> |:-:|:-:|:-:|:-:|:-:|:-:|:-:|:-:|:-:|:-:|
> |1|1e-8|1.0|1.0|68.0|11.2|10.2|89.7|23.8|10.0|
> |2|1e-8|0.3|0.3|62.6|10.8|10.0|90.3|24.5|9.9|
> |3|1e-8|0.1|0.1|**59.6**|**10.4**|**9.8**|**85.1**|**21.4**|**9.9**|
> |4|1e-7|0.1|0.1|63.5|10.7|10.0|88.8|24.2|10.0|

---

### Official Review · Reviewer_9ERg · 2023-10-30

**Soundness:** 2 fair
**Presentation:** 3 good
**Contribution:** 2 fair
**Rating:** 5
**Confidence:** 5

**Summary:**

This paper introduces the Multi-Modal Latent Diffusion Model (MM-LDM) for Sounding Video Generation (SVG). The main contributions of this paper are two-fold:
1. MM-LDM establishes audio and video latent spaces for SVG, which significantly reduces computational complexity.
2. MM-LDM proposes a multi-modal autoencoder to compress video and audio signals from pixel space to a semantically shared latent space.
The proposed method achieves state-of-the-art results, demonstrating its effectiveness.

**Strengths:**

1. The authors attempt to solve a novel and valuable problem and design a reasonable framework for this purpose.
2. The multimodal VAE designed by the author is interesting, establishing semantic latent spaces for audio and video modalities. Further, the authors use a shared multimodal decoder introduced in cross-modal alignment, which can inspire future multimodal generation.
3. The experimental results are promising in metrics, demonstrating the effectiveness of the proposed method. In particular, the MM-LDM achieves state-of-the-art results.

**Weaknesses:**

1. The designing of a multimodal VAE is innovative, but it may not be as effective as that of two separate VAEs. The authors should compare their multimodal VAE with most direct audio and video VAEs, which would better demonstrate the effectiveness of multimodal VAE.
2. I have viewed the generated results provided by the author, and only some results from the AIST++ dataset are available (MM-LDM: Multi-Modal Latent Diffusion Model for Sounding Video Generation (anonymouss765.github.io). However, the movements of the video characters are not natural, and due to the similarity of the generated audio, it is not clear whether the two modalities are in temporal alignment. Although the author has demonstrated significant success in metrics for their proposed method, it is necessary to supplement a human evaluation with MM-Diffusion to enhance credibility.
3. The paper only does experiments on small datasets, and there may be serious overfitting problems for diffusion-based methods. The author should discuss the generalization of their method on larger datasets.

**Questions:**

1. In Table 2, parts “Multi-Modal Generative Models on Audio-to-Video Generation” and “Multi-Modal Generative Models on Video-to-Audio Generation”, the results of MM-Diffusion don’t take the other modality as a condition input.
2. There is a typo in the heading of Table 3. “Latent Average Poolong” should be “Latent Average Pooling”.
3. On page 5, in the section “Signal Decoding”, the authors state that they initialize their signal decoder with parameters of a pre-trained image diffusion model to reduce training time and enhance the quality of reconstruction. It is confusing to initialize the decoder with a diffusion model as they are for different objectives. I wonder if this initialization has positive benefits.

---

> ### Author Response · Authors · 2023-11-17
> **Official Comments to Reviewer 9ERg**
>
> # Weakness #1: Comparison with Single-modal VAEs
> Thank you for your suggestion. As indicated in the ablation studies "Our base autoencoder independently decodes signals for each modality based on the respective spatial information from perceptual latents.", we have evaluated the performance of our multi-modal auto-encoder that removes all cross-modal correlations, which brings a significant performance drop. The only difference between the multi-modal auto-encoder used in the first ablation experiment with separate single-modal auto-encoders is that the signal decoder of two modalities are shared in our multi-modal auto-encoder due to computation limits. For further exploration, we separately train two single-modal auto-encoders under the same settings and report their results in the following table. It can be seen that the multi-modal auto-encoder outperforms two single-modal auto-encoders in all evaluation metrics at both $64^2$ and $256^2$, demonstrating the necessity of cross-modal correlations.
> |Model|FVD 64x|KVD 64x|FAD 64x|FVD 256x|KVD 256x|FAD 256x|
> |-|-|-|-|-|-|-|
> |Video AE|82.1|4.6|-|106.7|10.9|-|
> |Audio AE|-|-|9.1|-|-|9.1|
> |Multi-modal AE|**53.9**|**2.4**|**8.9**|**61.7**|**3.3**|**8.9**|
>
> # Weakness #2: Requiring Human Evaluation
> Synthesized videos on both AIST++ and Landscape datasets have been posted in the web (https://anonymouss765.github.io/MM-LDM). We have present the human evaluation results in the official comment Q2, please refer to it for details.
>
> # Weakness #3: Requiring Open-domain experiments
> Please refer to the official comments Q1.
>
> # Question #1: MM-Diffusion Cross-modal Generation Performance
> Thank you for bringing this issue to our attention. Based on the checkpoints released by MM-Diffusion, we re-measured its performance under V2A and A2V settings on the Landscape and AIST++ datasets. The results are reported in the following table, and we have modified the correlated part in our paper.  The cross-modal conditional generation results of MM-Diffusion are better than their single-modal generation results. Nonetheless, our MM-LDM still outperforms MM-Diffusion by a large margin, demonstrating the necessity of establishing perceptual latent spaces for high-resolution sounding video generation.
> |Method|Resolution||Landscape|||AIST++||
> |:-|:-:|:-:|:-:|:-:|:-:|:-:|:-:|
> |||FVD|KVD|FAD|FVD|KVD|FAD|
> |Audio-to-Video Generation||||||
> |MM-Diffusion*|64|237.9|13.9|-|163.1|28.9|-|
> |MM-Diffusion-SR*|64|225.4|13.3|-|142.9|24.9|-|
> |MM-LDM (ours)|64|**89.2**|**4.2**|-|**71.0**|**10.8**|-|
> |MM-Diffusion-SR*|256|347.9|27.8|-|225.1|51.9|-|
> |MM-LDM (ours)|256|**123.1**|**10.4**|-|**128.5**|**33.2**|-|
> |Video-to-Audio Generation||||||
> |MM-Diffusion*|-|-|-|9.6|-|-|12.6|
> |MM-LDM (ours)|-|-|-|**9.2**|-|-|**10.2**|
> # Question #2:
> Thank you for your comment. We have corrected this typo in our paper.
>
> # Question #3: The Necessity of Weight Initialization
> In this study, we reformulate the task of signal decoding as conditional generation, i.e. synthesizing video frames and audio images condition on their 2D perceptual latents. As depicted in Fig. 3(b), we employ the diffusion UNet in image diffusion methods to be our generative backbone. Based on this design, the objective of a diffusion model is integrated into the auto-encoder training loss, as in Equ. (3), where L_{MSE} is the noise mean square error loss used to train the diffusion model through \episilon-prediction. Initializing the generative model with a pre-trained image diffusion model significantly reduces the overall training times, as the initialized weights provide a robust and well-trained starting point of the signal decoder. To be more convincing, we conduct a simple comparison between auto-encoders with and without the weight initialization on the AIST++ dataset with a total of 16K steps, and report the results in the following table. It can be seen that the auto-encoder with weight initialization obtains significantly better decoding performance, demonstrating the necessity of weight initialization.
> ||FVD 64x|KVD 64x|FAD 64x|FVD 256x|KVD 256x|FAD 256x|
> |-|:-:|:-:|:-:|:-:|:-:|:-:|
> |w weight initialization|59.6|10.4|9.8|85.1|21.4|9.9|
> |wo weight initialization|129.8|24.5|10.3|182.5|42.9|10.3|

---

### Official Review · Reviewer_ApFF · 2023-10-31

**Soundness:** 3 good
**Presentation:** 3 good
**Contribution:** 3 good
**Rating:** 8
**Confidence:** 5

**Summary:**

In this paper, the author studies the problem of sounding video generation (SVG) and proposes a novel multi-modality video generation model in latent space named MM-LDM. The idea of incorporating the latent diffusion model and SVG task is interesting and the overall result is promising.

**Strengths:**

1. The idea of modeling audio and video in latent space for sounding video generation is interesting and promising.
2. The writing is good and the results further demonstrate the effectiveness of the proposed method.

**Weaknesses:**

1. Similar ideas to the conditional generation section have been proposed in many papers which seems too weak to list as a technical contribution in the paper. I would like the author to claim this point as a "bonus" of the proposed model in the paper.
2. The visual quality of MM-Diffusion results in Fig. 4 seems quite different from their original paper even considering the result has been super-resolved by the SR model. Is there any explanation for that? The visual quality of the results seems to be in low resolution or processed by some simple SR methods like nearest-neighbor.

**Questions:**

Please refer to the weakness part.

---

> ### Author Response · Authors · 2023-11-17
> **Official Comments for Reviewer ApFF**
>
> # Weakness #1: Reclaim the conditional generation approach
> Thank you for your valuable suggestion. We have modified our presentation of this approach in our newly uploaded PDF.
>
> # Weakness #2: Check Visual Samples of MM-Diffusion
> Thank you for bringing this issue to our attention! After carefully checking, we identified a flow in the released MM-Diffusion script. In particular, their two-stage synthesis script inconsistently saves mp4 videos from after super-resolution and png video frames from before super-resolution. This inconsistency impacts the visual quality of the results displayed in Fig. 4, since we directly paste those video frames when drawing Fig. 4. We sincerely apologize for our oversight, and we have rectified this figure in the current paper!

---

### Official Review · Reviewer_1VF7 · 2023-11-01

**Soundness:** 2 fair
**Presentation:** 1 poor
**Contribution:** 2 fair
**Rating:** 5
**Confidence:** 4

**Summary:**

The present paper proposes a framework based on the latent diffusion model to address the challenge of audio-visual joint generation. In comparison to the baseline (MM-Diffusion), the generation scheme proposed in this article, which operates in the latent space, offers significantly enhanced accuracy and reduces the computational burden. Moreover, the application of contrastive loss is also more judicious than in previous methods.
However, the author's writing exhibits some instances of ambiguity, which may suggest a lack of thorough understanding of the latent diffusion model. While the article's motivation and approach are commendable, I suggest that the author consider further revision and refinement of the manuscript before submitting it to the next conference.

**Strengths:**

1. The application of diffusion in latent space has been demonstrated in the fields of image and video generation, thereby warranting its extension to multi-modal generation. This approach to model generation is highly relevant in the current context of machine learning and artificial intelligence, where multi-modal data is increasingly prevalent. By leveraging the power of diffusion in latent space, multi-modal models can be developed that can generate diverse outputs across various modalities. This approach offers significant benefits in terms of model robustness, scalability, and generalization, making it an attractive choice for businesses and academic researchers alike.

2. The present study's results exhibit substantial enhancements from both qualitative and quantitative perspectives. The research outcomes demonstrate a significant improvement in both the quality and quantity of the study's output.

3. The proposed module in this article has been validated through extensive ablation experiments. The results of these experiments indicate the module's efficacy in addressing the intended objectives.

**Weaknesses:**

1. I think the author's understanding of latent diffusion is not deep enough, and there are many unprofessional and unscientific descriptions in the writing process. For details, see Questions 1, 2 and 4.

2. The sign of equation (7) is confusing. According to the paper, (n_a^t,n_v^t) are predicted noise features. But obviously this variable is not a predicted value, but a variable that satisfies the N(0,1) distribution.

3. The Implementation Details only give the training details of the multi-modal autoencoder, but not the training details of the diffusion model. Or is it that the model in this paper does not need to first train an autoencoder and then perform diffusion training, like [1]?

[1] Blattmann, Andreas, et al. "Align your latents: High-resolution video synthesis with latent diffusion models." Proceedings of the IEEE/CVF Conference on Computer Vision and Pattern Recognition. 2023.

**Questions:**

1. "we can leverage pre-trained image diffusion models to be our signal decoders". How is this step implemented? Are the "signal decoders" used here the diffusion unet in the image diffusion model? I don't understand how image diffusion model can act as decoder in autoencoder.

2. What is the relationship between the content drawn in Figure 3(b) and T? Why does the multi-modal autoencoder perform a denoising process?

3. In contrastive loss, how are positive and negative samples constructed, and how to define "matched pairs" during the implementation process?

4. "we utilize the ϵ-prediction to optimize our signal decoder, which involves the noise mean square error loss". What is the relationship between the process of training autoencoder and the method of noise prediction?

---

> ### Author Response · Authors · 2023-11-17
> **Official Comment for Reviewer 1VF7**
>
> # Weakness #1
> We are sorry for making you confused and mis-understanding our paper. We find the key divergence lies in how image diffusion models act as signal decoders. In particular, an image diffusion model can perform the signal decoding task (i.e. decoding signals from features) through conditional generation (i.e. synthesizing signals under strong conditions). A similar approach is also adopted in a contemporary work GLOBER [2], as specified in the "Video Decoding" paragraph: "We view the synthesis of video frames as a conditional diffusion-based generation task". Notably, our MM-LDM focuses on the audio-video multi-modal generation and proposes multiple strategies to overcome the inherent differences between the two modalities, which are different from GLOBER. We also provide detailed explanations for each question, please refer to Questions 1, 2, and 4 below.
>
> [2] Sun M, Wang W, Qin Z, et al. GLOBER: Coherent Non-autoregressive Video Generation via GLOBal Guided Video DecodER. NeurIPS 2023.
>
> # Weakness #2
> Thanks for your comment. As explained in Equ. (5), the terms $n_a^t$ and $n_v^t$ represent noise features sampled to corrupt the real audio and video signals during the forward diffusion process. Notably, these noise features serve as our training labels rather than outputs of the model. Conversely, $n_a(\*; \theta)$ and $n_v(\*; \theta)$ signify the predicted audio and video noise features generated by our multi-modal latent diffusion model, as specified in Equ. (7), where $\theta$ represents the modal parameters. We did not put the sign \theta as the subscript of n since the subscript of n is occupied by modality indicators such as a and v. Besides this, our notations align with previous works like Equ. (2) in LDM[3]. According to your comments, we will adopt $\tilde{n}^a_{\theta}(\*)$ and $\tilde{n}^v_{\theta}(\*)$ to denote the predicted audio and video noise features to eliminate confusion.
>
> [3] Rombach R, Blattmann A, Lorenz D, et al. High-resolution image synthesis with latent diffusion models. CVPR 2022.
>
> # Weakness #3
> Due to limited space, we have put more details about the training details and model settings for the multi-modal latent diffusion models (i.e. the Video Generator) in Appendix 1.4. This is indicated in the concluding sentence of the "Implementation Details" paragraph in Section 4.1. Please refer to the appendix for details.
>
> # Question #1
> Yes, the diffusion unet used in the image diffusion model is the main component of our signal decoder. The image diffusion model can perform the signal decoding task through conditional generation. To be specific, we reformulate the task of signal decoding (i.e. reconstruct the video frames and audio images based on their 2D perceptual latents) into synthesizing video frames and audio images condition on their 2D perceptual latents. To reduce stochasticity and obtain fidelity, the perceptual latent is mapped into multi-scale sub-features to provide strong guidance, as depicted in Figure 3(b). Moreover, the incorporation of other conditions such as modality and class embeddings also truncates the sampling space of the diffusion model and provides valuable instructions.
>
> # Question #2
> Figure 3(b) depicts the denoising process of signal decoders, which synthesize video and audio signals based on multiple conditions, and T denotes the total diffusion steps used in this process. As specified in question #1, our signal decoder reformulates the signal decoding task into conditional generation, and utilizes a diffusion unet as the generative backbone.
>
> # Question #3
> During each training step, we obtain a batch of sounding video clips with the batch size being B. Based on this batch of clips, we can obtain a batch of audio features $s_a^i, i=\{1,2,...,B\}$ and a batch of video features $s_v^i, i=\{1,2,...,B\}$. Obviously, $s_a^i$ and $s_v^i$ belong to the same clip, and $s_a^i$ and $s_v^j$, $i \neq j$, belong to different clips.
> As demonstrated in Equ. (1), we construct positive samples with paired audio-video features that belong to the same clip, and negative samples with unpaired audio-video features that belong to different clips. During implementation, audio-video features with the same batch indexes,  i.e. matched pairs,  are our positive samples.
>
> # Question #4
> As specified in Question #1, our signal decoder synthesizes video frames and audio images based on multiple conditions using a diffusion unet, where \epsilon-prediction loss is a common loss used to train diffusion models. As specified in Equ. (3), the \epsilon-prediction loss (i.e. L_MSE) is the primary component of the autoencoder training loss.

---

### Author Response · Authors · 2023-11-17
**Official Comments**

We sincerely thank all reviewers for their positive feedback that considers our work to be well written or easy to understand (Reviewer ApFF, 9ERg, and n6co), effective and promising (Reviewer 1VF7, ApFF, 9ERg, and n6co), and solve a valuable and challenging task (Reviewer 9ERg and n6co) as well as their advice. We have carefully considered the comments and addressed all the concerns in detail.

# Q1: Open-domain Generation Results
Limited by computational resources, we randomly select 10% samples (approximately 100K real videos) from the AudioSet dataset and train our MM-LDM for 5 days in total with 2 A800 gpus. The results are presented in Appendix 6.6 in the newly uploaded PDF, where all audios can be displayed using Adobe Acrobat PDF Reader by clicking the figures. Despite insufficient training, our MM-LDM can synthesize more diverse videos than MM-Diffusion with consistent audios, demonstrating the effectiveness of our proposed method.

# Q2: Human evaluation
For a thorough assessment, we conduct a manual evaluation of videos sampled by both MM-Diffusion and our MM-LDM on the landscape dataset. We randomly synthesize 500 sounding video samples using each model, obtaining a total of 1000 samples. These samples are then divided into five groups, with each group consisting of 200 samples for user rating. Following MM-Diffusion, each video is evaluated from three perspectives: audio quality (AQ), video quality (VQ), and audio-video matching (A-V), corresponding to sounding clarity, visual realism, and audio-video alignment, respectively. Each rating has a maximum score of 5. Each group of samples are shuffled and delivered to two users for voting. The average scores are shown in the table below. Our MM-LDM significantly surpasses MM-Diffusion by 0.52 AQ, 1.58 VQ, and 0.30 A-V, demonstrating the effectiveness of our proposed method and the necessity of constructing multi-modal latent space.

|Method|AQ$\uparrow$|VQ$\uparrow$|A-V$\uparrow$|
|-|-|-|-|
|MM-Diffusion|2.46|2.10|2.99|
|MM-LDM|**2.98**|**3.68**|**3.29**|

# Q3: Update results on Landscape
As specified in the "Training Targets" paragraph, KL losses are incorporated to punish the distributions of audio and video latent towards an isotropic Gaussian distribution. Considering that videos in Landscape are distinct nature scenes from diverse categories, the smaller KL loss allows more diversity of their latent features. We find that on the Landscape dataset, our MM-LDM performs much better with the KL loss weight being 1e-9 compared to 1e-8. To this end, we update our results and corresponding description on the Landscape with KL loss weight being 1e-9 in our new uploaded PDF file.

---

### Author Response · Authors · 2023-11-22
**Discussion period**

Dear reviewers 1VF7, ApFF, 9ERg, and n6co:

We are sorry to disturb you, while we have provided our responses for five days and there is not so much time left for the discussion stage. If you still have questions about our work, please let us know and we will reply as soon as possible.

Thanks for your effort and time!

Authors

---

### Meta-Review · Area_Chair_UvyA · 2023-12-06

**Metareview:**

The paper proposes a multi-modal latent diffusion model for audio-visual joint generation, demonstrating improvements in accuracy and computational efficiency over the baseline MM-Diffusion model. Reviewers raised several concerns, including: 1) the novelty of the approach is questioned, 2)  its effectiveness is not sufficiently validated on diverse or large datasets, and 3) limited number of examples provided for open-domain and long videos. Authors have provided their responses by adding more experiments and a user study, but reviewers did not change their rating. While the approach has some merits, the AC concurs with reviewers that the paper can be improved further in terms of contributions and experiments (focusing more on open-domain and long video generation).

**Justification For Why Not Higher Score:**

Reviewers pointed out that the paper lacks novelty, as its approach is seen as a straightforward extension of existing methods to latent space, which is a well-known strategy in diffusion-based generation. The limited scope of datasets used and the lack of comprehensive validation on diverse or large datasets prevent a higher evaluation.

**Justification For Why Not Lower Score:**

Despite its limitations, the paper addresses a challenging task and presents a method that shows potential in bridging the gap between audio and video modalities efficiently. The paper is well-presented and easy to understand.

---

### Decision · Program_Chairs · 2024-01-16

Reject